

# West Nile Virus (*Orthoflavivirus nilense*) RNA concentrations in wastewater solids at five wastewater treatment plants in the United States

Alessandro Zulli[1,2], Dorothea Duong[3], Bridgette Shelden[3], Amanda Bidwell[1], Marlene K. Wolfe[4], Bradley White[3] and Alexandria B. Boehm[1]

[1] Civil & Environmental Engineering, Stanford University, Stanford, CA, United States of America
[2] Division of Infectious Diseases and Geographic Medicine, Stanford University School of Medicine, Stanford, CA, United States of America
[3] Verily Life Sciences, South San Francisco, CA, United States of America
[4] Gangarosa Department of Environmental Health, Rollins School of Public Health, Emory, Emory University, Atlanta, United States of America

Corresponding author
Alexandria B. Boehm,
aboehm@stanford.edu

## ABSTRACT

**Background**. *Orthoflavivirus nilense*, formerly known as West Nile Virus (WNV), has become endemic to the United States since its introduction in 1999. Current surveillance methods rely primarily on mosquito pool testing, which is both costly and time-intensive. Wastewater-based epidemiology (WBE) has proven an effective method for the surveillance of various pathogens, including other orthoflaviviruses such as Dengue. WBE for WNV represents a potentially valuable surveillance approach that has so far been underexplored.

**Methods**. A targeted droplet digital reverse transcription-polymerase chain reaction (RT-PCR) approach (ddRT-PCR) was used to measure WNV concentrations in wastewater retrospectively from five locations and in over 600 samples. Three of these locations were in communities with multiple confirmed WNV infections, while two were not. Samples were collected during periods corresponding to typical WNV seasonality (spring to fall). SARS-CoV-2 RNA was measured simultaneously to assess nucleic acid degradation during sample storage. Publicly available confirmed WNV case data were compiled from the California and Nebraska departments of public health and their weekly arboviral reports.

**Results**. WNV RNA was detected in wastewater samples during periods of known viral circulation within a community. The adopted ddRT-PCR assay is highly specific and sensitive, and detections in wastewater solids correspond to the occurrence of cases in the season and location of sampling. WNV was detected in nine samples in three locations with known WNV clinical cases—wastewater positivity rates in these locations ranged from 3.3% to 13%. The results suggest that wastewater monitoring could serve as an effective complement to traditional surveillance methods, particularly for sentinel surveillance in locations which do not have extensive mosquito and clinical testing systems.

## INTRODUCTION

*Orthoflavivirus nilense*, formerly West Nile Virus (WNV), is a mosquito-borne single-stranded RNA virus first identified in Uganda in 1937 that has since been introduced to every continent except Antarctica (*Chancey et al., 2015*). The virus is spread in humans by the *Culex* mosquito, which acts as the transmission vector between birds and humans (*Chancey et al., 2015*; *World Health Organization, 2025*). Since Centers for Disease Control (CDC) surveillance of the virus began in 1999, the virus has caused an estimated 4.2 million infections, 59,141 confirmed cases, 27,617 hospitalizations, and 2,958 deaths (*CDC, 2025*). While an estimated 80% of cases are asymptomatic, approximately half of confirmed cases were neuroinvasive, often causing significant morbidity and early disability (*McDonald et al., 2021*; *Santini et al., 2022*; *CDC, 2025*). The impact of the virus in the United States is therefore significant, and the growing ecological range of the *Culex* mosquito due to climate change, amongst other factors, has led to West Nile virus becoming a significant public health concern (*Paz, 2015*; *Heidecke, Schettini & Rocklöv, 2023*; *Erazo et al., 2024*). Portions of this text were previously published as part of a preprint (*Zulli et al., 2025*).

Since its introduction in 1999, WNV has become endemic to the United States, consistently causing thousands of human cases each year, along with sporadic epizootics and avian/equine infections (*Ronca, Ruff & Murray, 2021*; *CDC, 2025*). Typically, human cases peak in late summer and early fall, and then rapidly decrease as temperatures drop below 18 °C, at which point the virus rarely establishes itself in *Culex* mosquitoes (*Di Pol, Crotta & Taylor, 2022*; *Vollans et al., 2024*). As there is no human vaccine available, surveillance and control methods primarily focus on mosquito pool surveillance, mosquito control, and public health education (*Ronca, Ruff & Murray, 2021*). Mosquito control includes various strategies, such as larval control, adult mosquito control, and public health surveillance. Many states, such as Florida and Nevada, regularly test mosquito pools for arboviruses during spring, summer and fall, allowing for granular mapping of the virus' prevalence (*Florida Department of Health, 2025*; *Southern Nevada Health District, 2025*). However, these methods are both costly and time-intensive, requiring dedicated teams and systems, and often providing epidemiological information with a significant delay.

Wastewater based epidemiology (WBE) has demonstrated to be an effective method for tracking and providing epidemiological information on various human pathogens including SARS-CoV-2, influenza, Chikungunya and Dengue virus (*Peccia et al., 2020*; *Boehm et al., 2023a*; *Monteiro et al., 2023*; *Wolfe et al., 2024*). In patients with WNV infections demonstrating neurological symptoms, over 50% of urine samples were found to be positive for WNV at higher relative concentrations than cerebrospinal fluid—50.5% of all urine samples compared to 2.8% of cerebrospinal fluid samples—indicating that the virus is likely shed into wastewater streams (*Gdoura et al., 2022*). Note that the authors did not provide concentrations in externally valid units; they provided cycle threshold (CT) values from a quantitative PCR instrument. Further, studies have shown that WNV preferentially partitions into wastewater solids (*Roldan-Hernandez, Van Oost & Boehm, 2024*), which are typically used for WBE. Given the wastewater detections of other flaviviruses such Dengue virus, known shedding of WNV in the urine of infected patients,

**Table 1  Primers and probes used in this study.**

| | | |
|---|---|---|
| WNV | Forward | TCAGCGATCTCTCCACCAAAG |
| | Reverse | GGGTCAGCACGTTTGTCATTG |
| | Probe | TGCCCGACCATGGGAGAAGCT |
| SARS-CoV-2 | Forward | CATTACGTTTGGTGGACCCT |
| | Reverse | CCTTGCCATGTTGAGTGAGA |
| | Probe | CGCGATCAAAACAACGTCGG |

and the difficulty in implementing environmental and clinical surveillance systems for WNV, WBE could provide a rapid, cost-effective method for gathering epidemiological information on entire communities (*Gyure, 2009*; *Barzon et al., 2013*; *Gdoura et al., 2022*; *Wolfe et al., 2024*). Despite the potential advantages of WBE for arboviral surveillance, there has been limited investigation into its application for WNV monitoring.

In this study, the viability of using wastewater to assess community infections of WNV is assessed. A targeted droplet digital RT-PCR approach was used to measure WNV concentrations in over 600 wastewater samples from five different wastewater treatment plants. These results demonstrate the potential promise and limitations of WBE for sentinel monitoring of the spread of WNV. Portions of this text were previously published as part of a preprint (*Zulli et al., 2025*).

# METHODS

## WNV assay

We used a previously published and validated hydrolysis-probe assay targeting the polyprotein gene of WNV (*Lanciotti et al., 2000*) in droplet digital RT-PCR (Table 1). Prior to our study, we carried out additional specificity and sensitivity testing *in silico* and *in vitro*. The assay was tested *in silico* for specificity by blasting the sequences in National Center for Biotechnology Information (NCBI) in October 2023. We first ran an inclusionary BLAST to confirm it matched recent WNV genomes, and then an exclusionary BLAST search excluding all WNV genomes, thereby only giving off-target matches. West Nile virus sequences downloaded from NCBI included RefSeq and 606 sequences collected between Jan 1 2020 and Jan 1 2024. Portions of this text were previously published as part of a preprint (*Zulli et al., 2025*).

The assay was tested *in vitro* for specificity and sensitivity using a virus panels (NATtrol Respiratory Verification Panel NATRVP2.1-BIO), and synthetic target nucleic acids (VR-3274SD) purchased from American Type Culture Collection (ATCC, Manassas, VA, USA), respectively. The respiratory virus panel includes chemically inactivated intact influenza viruses, parainfluenza viruses, adenovirus, rhinovirus A, metapneumovirus, rhinovirus, RSV, several coronaviruses, and SARS-CoV-2. Nucleic acids were extracted from intact viruses using Chemagic Viral DNA/RNA 300 Kit H96 for Chemagic 360 (PerkinElmer, Waltham, MA, USA). For *in vitro* testing, nucleic acids were used undiluted as template in digital droplet (RT-)PCR singleton assays for sensitivity and specificity testing in single wells. The concentration of targets used in the *in vitro* specificity testing was between $10^3$ and $10^4$ copies per well. Negative RT-PCR controls were included on each plate.

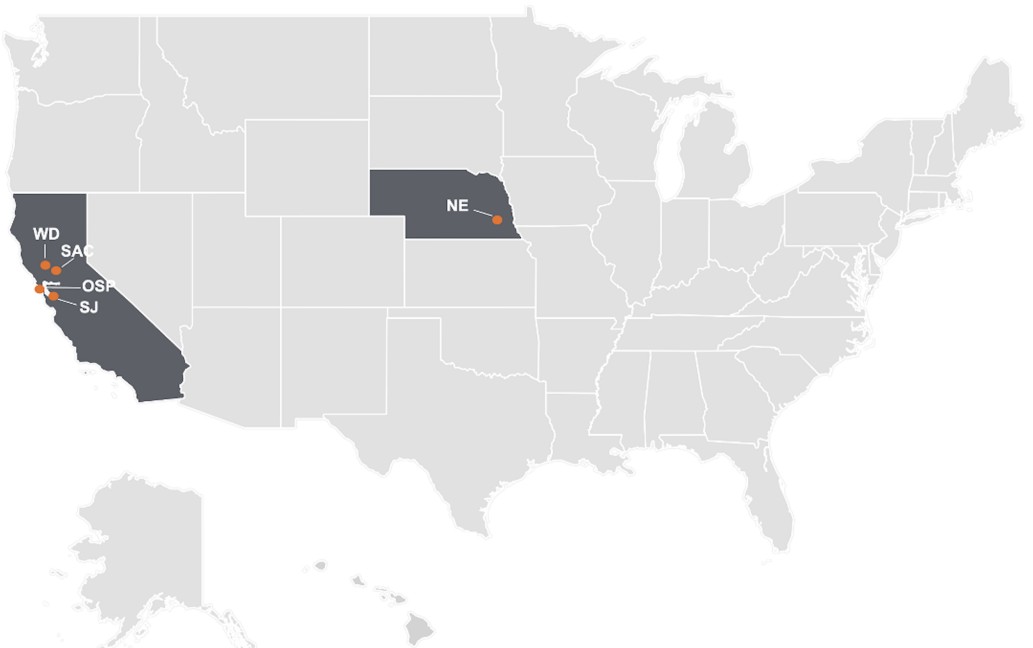

**Figure 1** **Map of the wastewater treatment plants (WWTPs) enrolled in this study.** The orange dots represent the location of the five WWTPs and states with participating WWTPs are shaded in dark gray. Generated using ArcGIS desktop; map layer from The United States Census Bureau's Cartographic Boundary Files (https://www.census.gov/geographies/mapping-files/time-series/geo/carto-boundary-file.html).

## Wastewater analyses

Wastewater samples from five different wastewater treatment plants (WWTPs) were tested for WNV RNA (Fig. 1). Two WWTPs, San Jose (SJ) and Oceanside (OSP), are located in the San Francisco Bay Area of California, USA and serve 1,500,000 and 250,000 people, respectively. We processed approximately two samples per week over a 26.5 month period (2/1/21–4/14/23, month/day/year format, $n = 229$ for SJ and $n = 230$ for OSP). SJ and OSP sites were chosen to represent sites where contributing communities were not expected to have WNV infections based on available case data (*California Department of Public Health, 2025*). Two WWTPs, Sacramento (SAC) and Woodland (WD), are located within or adjacent to Sacramento County, CA, USA and serve 1,480,000 and 59,000 people, respectively. Samples were collected approximately three times per week between 6/2/23 and 10/20/23 at SAC ($n = 60$) and between 6/2/23 and 10/11/23 at WD ($n = 51$). One WWTP was located in Lincoln, NE (NE) and serves 240,000 people; three samples were collected per week between 8/2/23 and 10/11/23 ($n = 31$). Different frequencies of sample analysis reflected different availability of samples for the project at the different sites. SAC, WD, and NE sites were included in the study because they had multiple confirmed WNV infections in the contributing populations during the time of sample collection (*Nebraska Department of Health Human Services, 2025*). Additional details of the sites can be found elsewhere (*Boehm et al., 2024*).

At SJ, SAC, and OSP, 50 mL of settled solids from the primary clarifier, and at WD and NE, 50 mL of raw influent sewage were collected; sterile technique and clean bottles were used in all cases. Samples from NE and WD were 24 h composites of influent, while samples from SJ, OSP, and SAC were "grab" samples from the primary clarifier. Samples were stored at 4 °C, transported to the lab, and processed within 48 h of collection, which was the quickest possible given needed transport; previous work suggests limited degradation of RNA targets over such a duration when stored at 4 °C (*Simpson et al., 2021*; *Zhang, Roldan-Hernandez & Boehm, 2024*). Wastewater solids were isolated from the raw influent using centrifugation, as described previously (*Boehm et al., 2024*). Then settled solids from all samples were dewatered (*Topol et al., 2021a*). At this point in processing (prior to nucleic-acid extraction), samples from SJ and OSP were frozen at −80 °C for 4–60 weeks, and then thawed overnight at 4 °C prior to further processing. Remaining samples were not frozen before nucleic-acid extraction.

Nucleic acids were obtained from the dewatered solids following previously published protocols and commercially available kits: Chemagic Viral DNA/RNA 300 Kit H96 (PerkinElmer, Shelton, CT) followed by inhibition removal (Zymo OneStep PCR Inhibitor Removal Kit, Irvine, CA, USA) (*Topol et al., 2021b*; *Boehm et al., 2023b*). Those protocols include suspending the solids in a buffer at a concentration of 75 mg/ml and using an inhibitor removal kit; together these processes alleviate potential inhibition while maintaining good assay sensitivity (*Huisman et al., 2022*; *Boehm et al., 2023b*). Nucleic-acids were obtained from six or 10 replicate sample aliquots (six replicates for SAC, NE, and WD; and 10 replicates for OSP and SJ). Each replicate nucleic-acid extract from each sample was subsequently stored between 8 and 273 days (median 266 days) for SJ, 1 and 8 days (median 5 days) for OSP, 14 and 154 days (median 84 days) for SAC, 32 and 163 days (median 98 days) for WD, and 32 and 102 days (median 67 days) for NE at −80 °C and subjected to a single freeze thaw cycle. Upon thawing, WNV and SARS-CoV-2 RNA were measured immediately using multiplex digital droplet RT-PCR. SARS-CoV-2 RNA was used as a storage control as it was measured previously in the samples without any storage (methods and results reported in a Data Descriptor (*Boehm et al., 2024*).

The WNV and SARS-CoV-2 N gene assay were run in multiplex using a probe-mixing approach and unique fluorescent molecules (HEX, FAM, Cy5, Cy5.5, ROX, and/or ATTO950). Nucleic-acid targets from from SJ and OSP were measured in multiplex for SARS-CoV-2 (fluorescent molecule(s) on probe: FAM/HEX) and WNV RNA (ROX) along with assays for three other human viral targets not reported herein (rotavirus (FAM), human adenovirus group F (ROX/ATTO590), human norovirus GII (ATTO590)). Nucleic-acid targets from SAC, WD, and NE samples were measured in a duplex assay for SARS-CoV-2 (HEX) and WNV (FAM) RNA. Each nucleic acid extract was run in a single well so that six or 10 replicate wells, depending on the site, were run for each sample for each assay.

Each 96-well PCR plate of wastewater samples included PCR positive controls for each target assayed on the plate in 1 well, PCR negative no template controls in two wells, and extraction negative controls (consisting of water and lysis buffer) in two wells. PCR positive

controls consisted of synthetic viral WNV gRNA (VR-3274SD, ATCC) and SARS-CoV-2 gRNA (VR-1986D, ATCC).

ddRT-PCR was performed on 20 μl samples from a 22 μl reaction volume, prepared using 5.5 μl template, mixed with 5.5 μl of One-Step RT-ddPCR Advanced Kit for Probes (1863021; Bio-Rad, Hercules, CA, USA), 2.2 μl of 200 U/μl Reverse Transcriptase, 1.1 μl of 300 mM dithiothreitol (DDT) and primers and probes mixtures at a final concentration of 900 nM and 250 nM respectively. Primer and probes for assays were purchased from Integrated DNA Technologies (IDT, San Diego, CA) (Table 1). WNV and SARS-CoV-2 RNA was measured in reactions with undiluted template. Primers and probes are provided in Table 1.

Droplets were generated using the AutoDG Automated Droplet Generator (Bio-Rad, Hercules, CA, USA). PCR was performed using the Mastercycler Pro (Eppendforf, Enfield, CT, USA) with the following cycling conditions: reverse transcription at 50 °C for 60 min, enzyme activation at 95 °C for 5 min, 40 cycles of denaturation at 95 °C for 30 s and annealing and extension at 59 °C (for WNV and SARS-CoV-2) for 30 s, enzyme deactivation at 98 °C for 10 min then an indefinite hold at 4 °C. The ramp rate for temperature changes were set to 2 °C/second and the final hold at 4 °C was performed for a minimum of 30 min to allow the droplets to stabilize. Droplets were analyzed using the QX200 or the QX600 Droplet Reader (Bio-Rad). A well had to have over 10,000 droplets for inclusion in the analysis. All liquid transfers were performed using the Agilent Bravo (Agilent Technologies, Santa Clara, CA, USA).

Thresholding was done using QuantaSoft™ Analysis Pro Software (Bio-Rad, Hercules, CA, USA, version 1.0.596) and QX Manager Software (Bio-Rad, version 2.0). Replicate wells were merged for analysis of each sample. In order for a sample to be recorded as positive, it had to have at least three positive droplets across all merged wells. We chose three positive droplets as a threshold for positive samples to avoid false positives.

Concentrations of RNA targets were converted to concentrations in units of copies (cp)/g dry weight using dimensional analysis. The total error is reported as standard deviations and includes the errors associated with the Poisson distribution and the variability among the six or 10 replicates. Three positive droplets across six to 10 merged wells corresponds to a concentration between ~500–1,000 cp/g; the range in values is a result of the range in the equivalent mass of dry solids added to the wells. Wastewater measurements are available from the Stanford Digital Repository (https://purl.stanford.edu/pp102gy1970).

## Case data

For the SJ, OSP, SAC, and WD locations, clinical case data was compiled at the county level from the Vector-Borne Disease Section of the California Department of Public Health, which publishes weekly data on detections in humans and animals of West Nile Virus and St. Louis encephalitis virus (*California Department of Public Health, 2025*). For the NE location, clinical case data for Nebraska was compiled from the Nebraska Department of Health and Human Services, which publishes weekly mosquito borne disease reports that include WNV for the entire state (*Nebraska Department of Health Human Services, 2025*). Clinical case data was collected for the same time periods as wastewater surveillance (SJ,

OSP: 2/1/21–4/14/23, WD, SAC: 6/2/23–10/20/23, NE: 8/2/23–10/11/23). All case data are publicly available.

## Statistics

Case data were normalized by the populations of the counties or states included in the case reporting to calculate incidence rates. As case data are available for morbidity and mortality weekly report (MMWR) weeks, variables were created to indicate whether WNV RNA was detected each MMWR week for each WWTP. We calculated Kendall's tau between WNV RNA detection each week (a binary variable) and the incidence rate of WNV infections in the contributing community to test the null hypothesis that there is no association. We also combined all the WWTP WNV RNA and incidence rate data together into one data set to test the same null hypothesis. Finally, the positivity rate was calculated across all samples from each of the five WWTPs, and the incidence rate of WNV infections across the entire time period each WWTP was studied and we tested the null hypothesis that there is no correlation across the two variables. We used Kendall's tau because the independent variables are not normally distributed. We used a $p$ value less than 0.05 to reject the null hypothesis.

# RESULTS

## WNV assay specificity and sensitivity

We used a previously published assay for WNV (*Lanciotti et al., 2000*) and that study conducted thorough sensitivity and specificity assessments. In order to ensure that the assay remained sensitive and specific, both *in silico* and *in vitro* analyses were carried out. Both analyses indicated that the WNV assay was specific and sensitive. The assay matched downloaded WNV sequences from a global database (NCBI) suggesting the assay is specific, and there was no cross reactivity with non-target sequences *in vitro*. The results from this testing, along with the extensive testing conducted by *Lanciotti et al. (2000)* supports the specificity and sensitivity of the assay. Portions of this text were previously published as part of a preprint (*Zulli et al., 2025*).

## QA/QC

All positive and negative controls were positive and negative, respectively. In a previous study, RNA recovery from the samples is reported, as inferred from recovery of a spiked bovine coronavirus, to be close to 1 (*Boehm et al., 2024*), and those results are not repeated herein. Median ratio of SARS-CoV-2 N gene measurements made in this study to those made using fresh samples was 0.2 at SJ and 0.4 at OSP suggesting the storage of wastewater solids and subsequent freeze thaw may have reduced measurement concentrations, but by less than an order of magnitude. Median ratio was 1.5 at NE, 1.6 at WD, and 1.3 at SAC suggesting that the storage of nucleic-acids did not reduce quantification and may have slightly enhanced it. Recall that storage of samples from the latter three WWTPs just involved storage of nucleic-acids, whereas storage of samples from SJ and OSP involved storage of wastewater solids and nucleic-acids. Although not ideal, storage of samples is essential for retrospective work like this.

Additional QA/QC details and reporting that follows those recommend by the MIQE (*The dMIQE Group & Huggett, 2020*) and Environmental Microbiology Minimal Information (EMMI) guidelines (*Borchardt et al., 2021*) are provided at the Stanford Digital Repository (https://purl.stanford.edu/pp102gy1970).

### WNV in wastewater

We measured WNV RNA in 601 samples across five WWTPs. The vast majority ($n = 592/601$, 98%) of the samples were negative for WNV RNA meaning the concentration was less than the lowest measurable concentration (approximately 1,000 cp/g). Nine samples (2%) were positive for WNV RNA with concentrations ranging from 1,755 cp/g to 11,404 cp/g (median = 7,146 cp/g). Positive samples were only observed at three of the sites: SAC ($n = 2$ positive samples), WD ($n = 3$ positive samples), and NE ($n = 4$ positive samples). No samples were positive at SJ or OSP. Positivity rates (number positive / total samples run) for SAC, WD, and NE are 3.3%, 5.9%, and 13%, respectively.

### Confirmed WNV cases

Weekly WNV case data are available aggregated across each county for California. In SAC, WD, SJ, and OSP, there were 29, 22, 1, and 1 cases respectively in the corresponding counties. Normalized by the county population, that is an incidence rate of 18, 100, 1, and 1 out of one million people for SAC, WD, SJ and OSP, respectively. Weekly WNV case data are available aggregated at the state for NE. There were 134 cases across the state, or an incidence rate of 67 out of one million people. Data for SAC, WD, and NE are provided in Fig. 2; cases for SJ and OSP were recorded for the week of 10/7/22 and 10/29/21, respectively.

### Relationship between WNV RNA in wastewater and WNV cases

For each of the three WWTPs where WNV RNA was detected, there is no association between the weekly presence of WNV RNA and the WNV infection incidence rate (tau = $-0.17$, 0.20, and 0.24; $p = 0.42$, 0.34, and 0.42; $n = 21$, 21, and 10 for SAC, WD, and NE, respectively). When data from all five sites were combined, there is a significant association between weekly presence of WNV RNA and incidence rate (tau = 0.33, $p = 2.6 \times 10^{-8}$, $n = 282$). The wastewater positivity rate and county or state-specific (for NE only) incidence rate, over the entire duration of the study, were positively correlated, but the correlation was not statistically significant (tau = 0.74, $p = 0.077$, $n = 5$) (Fig. 3).

## DISCUSSION

WNV RNA was detected in wastewater solids of three of the five WWTPs in this study. Detection of the viral RNA was not expected, and not observed, in two of the WWTPs where the most samples were analyzed; these WWTPs are located in a highly urbanized area in the San Francisco Bay Area of California where WNV infections are rare, and mosquito control programs are in place to control mosquito populations. On the other hand, WNV was detected at the three other WWTPs, with detection rates ranging from 3.3% (SAC) to 13% (NE). These three WWTPs were located in areas where WNV infections occur, and

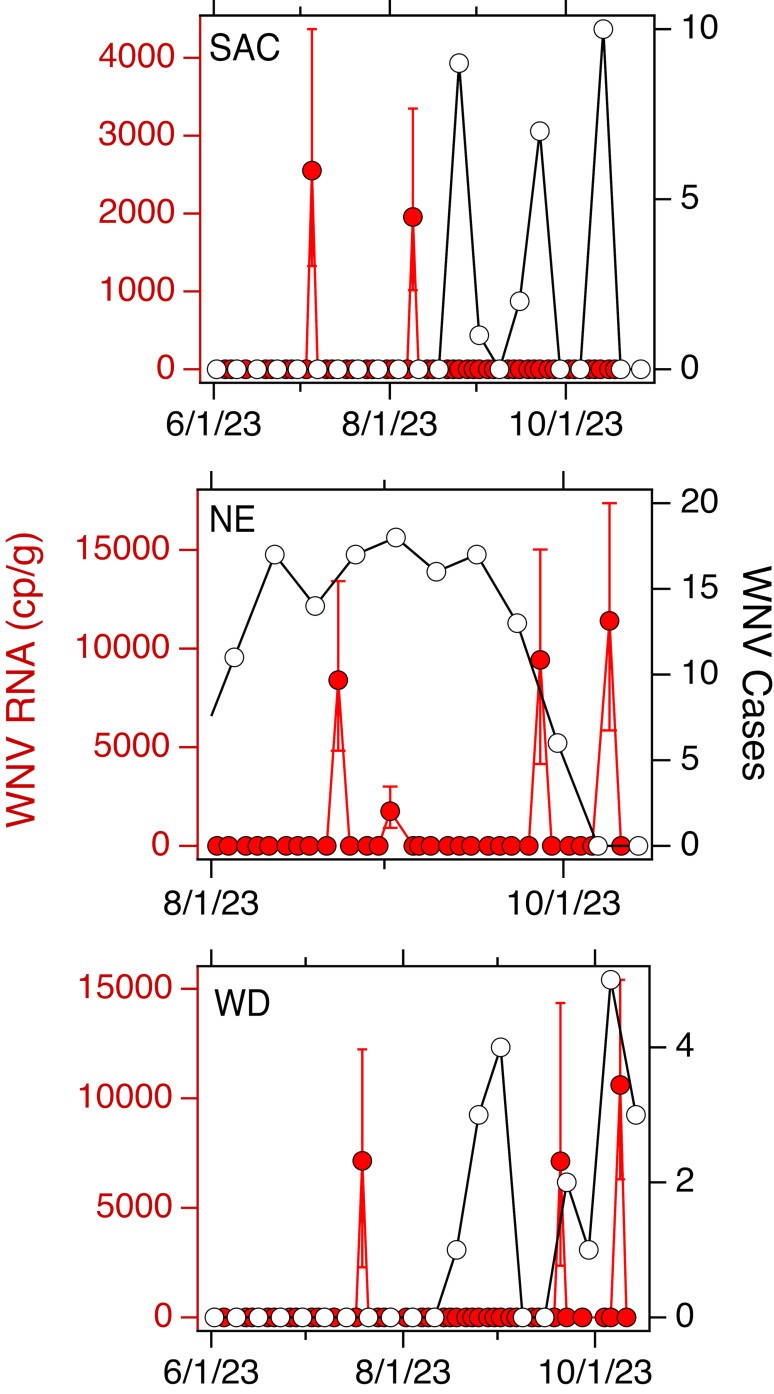

**Figure 2   WNV RNA and case data used in the study.** Concentrations of WNV RNA measured at SAC, NE, and WD sites during the study (in red, left axis) and confirmed cases of WNV infection in the associated county (SAC and WD) or state (NE) (in black, open circles, right axis). Error bars on the WNV RNA measurements represent standard deviations. A 0 value for concentration was imputed for samples where WNV RNA was not detected.

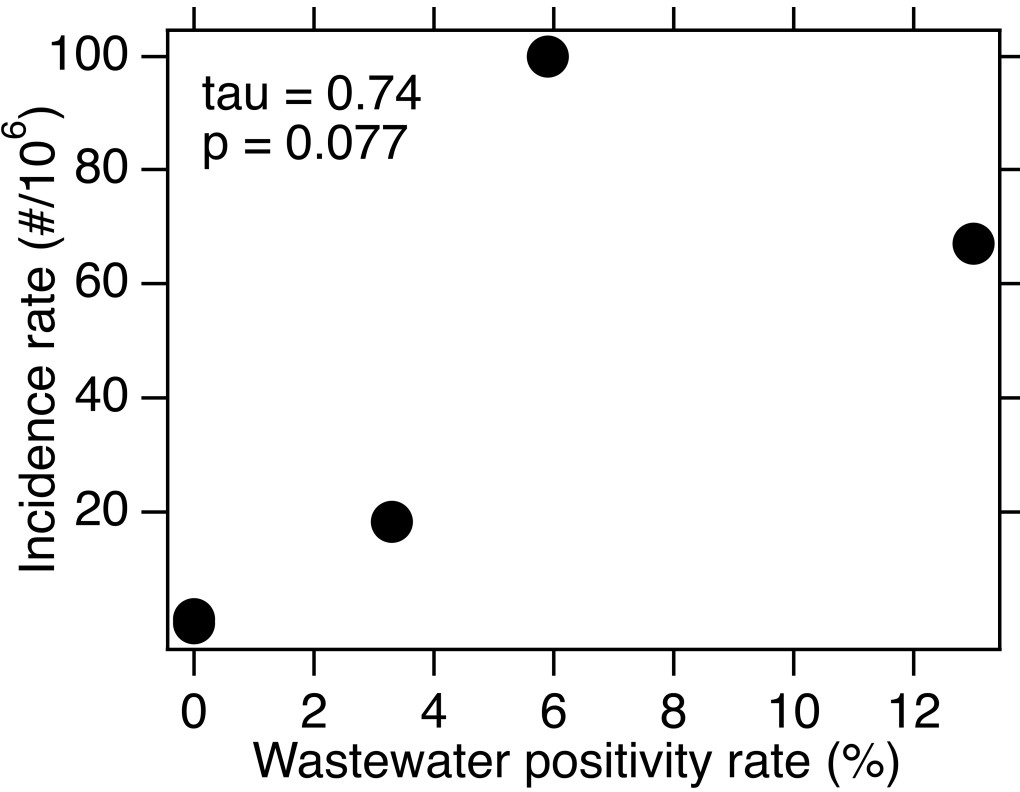

**Figure 3** **WNV incidence rate as a function of wastewater positivity rate.** Data is for the five sites over the time periods included for each in this study. The points representing SJ and OSP are on top of each other near the origin. Kendall's tau and associated *p* value are provided.

during times when WNV infections were recorded. Variations in sampling methods and frequency between some of the WWTPs might have impacted the results. Portions of this text were previously published as part of a preprint (*Zulli et al., 2025*).

Although within each WWTP there is no association between detection of WNV RNA and incident cases of WNV, an aggregated analysis demonstrated that weekly presence of WNV RNA was significantly correlated to incidence rate. Therefore, detection of WNV RNA is suggestive of WNV cases in the community contributing to the WWTP during that WNV season, and could potentially be used as a supplementary or low-cost method for monitoring WNV in low resource settings and the increasing ecological range of WNV (*Paz, 2015*; *Di Pol, Crotta & Taylor, 2022*; *Heidecke, Schettini & Rocklöv, 2023*). This finding adds to the existing evidence that WBE may be a useful tool in tracking and combating the spread of arboviruses.

Previous implementations of WBE for arbovirus detection have yielded similar results to the ones presented in this study, showing sporadic detections of viruses in endemic areas and during acute epidemics (*Fanok et al., 2023*; *Wolfe et al., 2024*; *Monteiro et al., 2025*). These viruses include Dengue, Chikungunya, and Japanese Encephalitis, with each showing sporadic detection patterns during outbreaks in the underlying community. Wolfe et al. demonstrated that even in areas with very low prevalence of Dengue (Miami),

detection was feasible, with 21% of samples positive for Dengue during a period with known clinical cases. Similarly, Monteiro et al. showed that in Portugal, where rates of Dengue and Chikungunya are low, Dengue was detected in 25% of samples, and Chikungunya in 11%. These demonstrate that arboviruses, and specifically flaviviruses (which include Dengue and WNV), are detectable through WBE in a manner consistent with underlying human transmission. *Fanok et al. (2023)* then showed that during an outbreak of Japanese Encephalitis Virus (JEV), 33% of samples tested had detectable levels of the virus. This finding is particularly relevant as JEV is a flavivirus closely related to WNV for which humans are also dead-end hosts. Finally, a pre-print by *Kuhn et al. (2025)* showed that WNV is sporadically detected in wastewater in Oklahoma, reinforcing this pattern. More specifically, *Kuhn et al. (2025)* detected WNV in four WWTPs across three counties, though no significant correlation to clinical or animal cases was found, leaving open the question as to whether detections of WNV in wastewater reflect the underlying human epidemiological situation. The results of the present study now demonstrate that these detections are not only possible and seasonal, but that the presence of WNV in wastewater can be indicative of clinical cases in humans, an important step in the implementation process of WBE for WNV.

WBE for arboviruses can be an important tool in combating the spread of arboviruses, particularly as climate change is rapidly changing and expanding the ecological ranges of the vectors that carry arboviral disease, including the *Culex* mosquitoes responsible for WNV (*Gilbert, 2021*; *Heidecke, Schettini & Rocklöv, 2023*; *Erazo et al., 2024*). As average temperatures rise in certain locations, many will cross the 18 °C threshold at which WNV establishes itself in mosquitoes. These increases in temperature will also lead to higher rates of human transmission as vectors and their feeding patterns increase (*Paz, 2015*). This means that geographic areas that do not regularly see endemic transmission of WNV virus will be faced with the challenge of rapidly establishing clinical and environmental systems for control such as the ones present in California, Florida, and Nebraska (*California Department of Public Health, 2025*; *Florida Department of Health, 2025*; *Nebraska Department of Health Human Services, 2025*). These systems can be capital and time-intensive, but WBE provides a potential alternative—a low-cost, sentinel surveillance system that can build upon an established network for monitoring other viruses such as SARS-CoV-2 and influenza. Our results demonstrate that WNV WBE is a relatively simple addition to existing systems that can provide real-time information on the spread of these viruses in human populations, allowing public health officials to react accordingly.

There are limitations associated with this work. The available case data are limited in that they rely on confirmed, symptomatic, and likely severe WNV infections with neurological complications. WNV infections can be mild or asymptomatic, and such cases would not be counted by the current health surveillance systems (*Santini et al., 2022*; *World Health Organization, 2025*). Several weeks in Nebraska and Sacramento showed no detections of WNV despite known clinical cases. In Nebraska, as the clinical data are only available at the state level, reported cases could have been outside the sewershed. In the case of Sacramento, the treatment plant serves nearly 1.5 million people with a limited number of known cases ($<= 10$), which could significantly dilute inputs from infected individuals.

Further, knowledge on the shedding of WNV is highly limited, and is currently only studied in the case of neuroinvasive presentation of the disease. There are limited data on WNV RNA shedding *via* human excretions, particularly given the context that humans are a dead end host for WNV. If shedding is low, then concentrations expected in wastewater will also be low. Although wastewater solids can provide sensitive detection of rare nucleic-acid targets owing to their natural ability to concentrate viral biomarkers, including WNV RNA *via* adsorption (*Roldan-Hernandez, Van Oost & Boehm, 2024*), perhaps an even more sensitive approach will be needed to measure expected low levels of WNV RNA in wastewater. Finally, some samples processed in this study underwent up to one freeze-thaw cycle, which has been shown to affect RNA concentrations of other viruses (*Simpson et al., 2021*). Storage and biobanking of samples is essential for research and future studies should continue to investigate how storage affects RNA and DNA quantification in wastewater.

## CONCLUSION

As the ecological range of the Culex mosquito continues to increase and increases the risk of WNV transmission in new geographic areas, wastewater-based epidemiology offers a cost-effective surveillance approach that can leverage pre-existing infrastructure. While challenges remain, including limited understanding of viral shedding from humans as dead end hosts, this study provides important evidence that wastewater-based epidemiology can serve as a viable sentinel surveillance system for WNV. By integrating this approach into existing WNV monitoring strategies, public health officials may gain additional context necessary to implement targeted interventions and mitigate the impact of WNV in both historically and newly endemic communities.

## ACKNOWLEDGEMENTS

We thank the participating wastewater treatment plants for their samples for the project.

### Funding
The work was supported by a gift from the Sergey Brin Family Foundation to Alexandria B. Boehm. The funders had no role in study design, data collection and analysis, decision to publish, or preparation of the manuscript.

### Grant Disclosures
The following grant information was disclosed by the authors:
The Sergey Brin Family Foundation.

### Competing Interests
Dorothea Duong and Bridgette Shelden are employees of Verily Life Sciences. Bradley White was an employee of Verily Life Sciences when this work was performed.

## Author Contributions

- Alessandro Zulli conceived and designed the experiments, performed the experiments, analyzed the data, prepared figures and/or tables, authored or reviewed drafts of the article, and approved the final draft.
- Dorothea Duong performed the experiments, analyzed the data, authored or reviewed drafts of the article, and approved the final draft.
- Bridgette Shelden performed the experiments, analyzed the data, authored or reviewed drafts of the article, and approved the final draft.
- Amanda Bidwell analyzed the data, prepared figures and/or tables, authored or reviewed drafts of the article, and approved the final draft.
- Marlene K. Wolfe conceived and designed the experiments, authored or reviewed drafts of the article, and approved the final draft.
- Bradley White conceived and designed the experiments, authored or reviewed drafts of the article, and approved the final draft.
- Alexandria B. Boehm conceived and designed the experiments, performed the experiments, analyzed the data, authored or reviewed drafts of the article, and approved the final draft.

## Data Availability

The data is available at Stanford Digital Repository: Boehm, A., Zulli, A., Duong, D., Shelden, B., White, B., Wolfe, M., and Bidwell, A. (2025). Data on West Nile Virus RNA in five wastewater treatment plants. Version 3. Stanford Digital Repository. Available at https://purl.stanford.edu/pp102gy1970/version/3. https://doi.org/10.25740/pp102gy1970.

## Supplemental Information

Supplemental information for this article can be found online at http://dx.doi.org/10.7717/peerj.19748#supplemental-information.

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
