# Peer review of "West Nile Virus (*Orthoflavivirus nilense*) RNA concentrations in wastewater solids at five wastewater treatment plants in the United States"

_PeerJ, doi:10.7717/peerj.19748_

## Round 0.1 · original submission · Major Revisions

The manuscript is well-prepared and presents valuable findings, but substantial revisions are required before it can be considered further. Particular attention should be given to clarifying methodological details like controls, sampling strategy, ddPCR thresholds, expanding assay performance data, and addressing concerns about storage duration and sample integrity. Revisions to Table 1 and the inclusion of a summary figure showing all sites are recommended. The abstract should be shortened to avoid redundancy with the introduction, and excessive use of personal pronouns should be minimized. Strengthening the discussion of discrepancies between clinical and wastewater data and reducing redundancy, especially regarding climate impacts, will improve the clarity and impact of the work.

Please revise the manuscript accordingly taking care to respond to each comment in a detailed response letter (minor stylistic suggestions and reference formatting can be addressed more briefly). Highlight changes in the revised manuscript for ease of review.

I look forward to receiving your revised version

·

Basic reporting

The manuscript is exceptionally well-written, with comprehensive details provided in the methods section. The conclusions are robustly supported by the data presented, and the authors have thoroughly addressed the study's weaknesses.

Experimental design

Although the Methods are clearly detailed, there are a few sections that require additional clarification:
L137: what type of samples were used as positive controls in the in vitro testing?
L162: how were the influent samples collected (composite or grab samples) ?
L228: was the threshold of 3 droplets applied across all replicates together? Can you provide justification on the chosen threshold?
Was any data available on flow rates to try normalize the wastewater data?

The Results section should include more specific information on the specificity and sensitivity of the assay used, both in silico and in vitro analyses. For lines 266-272, provide detailed results such as the number of positive and negative controls tested, Ct values, etc. Additionally, include information on the standard curve calibration for the ddPCR assay. For line 270, specify the number of sequences downloaded from NCBI.
Table 1: add Ta, size of the amplicon (bp), references, type and dye of probes.

Validity of the findings

Figure 2: Could you include a graph that combines data from all five sites, highlighting the significant association between WNV RNA and clinical cases? While the detection of WNV RNA in the absence of reported cases is understandable due to the monitoring and reporting methods, the absence of WNV RNA detection in wastewater despite the presence of cases in the watershed is concerning. Could you elaborate on this issue in the Discussion section?

Reviewer 2 ·

Basic reporting

Basic reporting
Can you reduce the use of personal pronoun such as “we”, “I”, or they in scientific writings.
Please reduce the abstract. It is rather too long. The background should be brief and not a duplicate of the introduction.
Line 90 is missing a period, before “How”.

Experimental design

Experimental design
In SARS-CoV-2 RNA analysis there were evidence that frozen and thawing have effects on the concentrations of RNA. Also, the frozen temperature and time have effects on the concentrations of RNA. Did you conduct any of such tests? I know you did single frozen – thaw cycle.
Is there any reason why the samples frequency varies from station to station. The minimum sampling frequency per week should be 3 times for monitoring trends.
Processing samples within 24 hours of collection is known to be the best for optimal results. Is there any reason for waiting 48 hours? Can you provide references for articles that process within 48 hours?
Did you use commercial extraction kit or in-house method?
Could the lack of correlation between clinical cases and wastewater data be related to sampling frequency, and storage time?
Line 347, what is Wolfe et al (ref)? I think the year is missing. Also check Monteiro et al., in line 349.
Kohn et al should have been published by now or it has been rejected, kindly look for the published version.
Line 360 – 365; I think both current results and Kohn results both did not show any association with clinical cases.

Validity of the findings

Validity of the findings
Only nine positive samples out of 601 could mean that the method is working, but lack of frequent detection even in places where positive clinical cases have been detected is worrisome. It could signal a problem with the protocol.

Additional comments

General comments
There are redundancies in the document, so authors are encouraged to remove them. For example, changes in climate change were overstated.
The conclusion is writing like a summary, please try to rewrite it.

---

## Round 0.2 · Minor Revisions

Thank you for your revised manuscript. The reviewer is pleased with the improvements and considers the manuscript to be in good shape. However, a few minor comments remain that should be addressed before final acceptance.

I kindly ask you to revise the manuscript accordingly and submit a final version incorporating these minor changes. Once these are addressed, I will proceed with the acceptance of your manuscript.

Best regards,

Reviewer 2 ·

Basic reporting

The manuscript has improved significantly.

Experimental design

Kindly mention the name of the commercial kit used. Thank you for providing citations in line 190.

Validity of the findings

Thank you for changing from "is" to "can be" in line 386. This particularly important since there is no correlation between positive wastewater data and incidence rate. Another common concern is underreporting of incidence rate due to infected patient not going for test or self-management of cases.

Could the differences in the sampling approaches affect the wastewater positivity?

Additional comments

Can you try to include effective reproduction numbers, training your data set with few available clinical cases, then used that to predict what will happen when no clinical cases are available. This will help policy make to make informed decision from wastewater data.

Calculating effective wastewater reproduction number will make your work stand out.

---

## Round 0.3 · accepted · Accept

Thank you for your revised submission and for addressing the reviewers’ comments. The modifications have improved the overall clarity and robustness of the manuscript. Your rationale for not including the reproduction number analysis is reasonable and scientifically sound. The manuscript is now suitable for publication.

I am pleased to inform you that your paper is accepted for publication.

Congratulations, and thank you again for submitting your work to PeerJ